

# Neighborhood structure-guided brain functional networks estimation for mild cognitive impairment identification

Lizhong Liang[1], Zijian Zhu[2], Hui Su[3], Tianming Zhao[4] and Yao Lu[1]

[1] School of Computer Science and Engineering, Sun Yat-sen University, Guangzhou, China
[2] School of Public Health, Guangdong Medical University, Dongguan, China
[3] Shandong Liaocheng Intelligent Vocational Technical School, Liaocheng, China
[4] Dalian University, Dalian, China

## ABSTRACT

The adoption and growth of functional magnetic resonance imaging (fMRI) technology, especially through the use of Pearson's correlation (PC) for constructing brain functional networks (BFN), has significantly advanced brain disease diagnostics by uncovering the brain's operational mechanisms and offering biomarkers for early detection. However, the PC always tends to make for a dense BFN, which violates the biological prior. Therefore, in practice, researchers use hard-threshold to remove weak connection edges or introduce $l_1$-norm as a regularization term to obtain sparse BFNs. However, these approaches neglect the spatial neighborhood information between regions of interest (ROIs), and ROI with closer distances has higher connectivity prospects than ROI with farther distances due to the principle of simple wiring costs in resent studies. Thus, we propose a neighborhood structure-guided BFN estimation method in this article. In detail, we figure the ROIs' Euclidean distances and sort them. Then, we apply the K-nearest neighbor (KNN) to find out the top K neighbors closest to the current ROIs, where each ROI's K neighbors are independent of each other. We establish the connection relationship between the ROIs and these K neighbors and construct the global topology adjacency matrix according to the binary network. Connect ROI nodes with k nearest neighbors using edges to generate an adjacency graph, forming an adjacency matrix. Based on adjacency matrix, PC calculates the correlation coefficient between ROIs connected by edges, and generates the BFN. With the purpose of evaluating the performance of the introduced method, we utilize the estimated BFN for distinguishing individuals with mild cognitive impairment (MCI) from the healthy ones. Experimental outcomes imply this method attains better classification performance than the baselines. Additionally, we compared it with the most commonly used time series methods in deep learning. Results of the performance of K-nearest neighbor-Pearson's correlation (K-PC) has some advantage over deep learning.

Corresponding author
Yao Lu, luyao23@mail.sysu.edu.cn

## INTRODUCTION

Alzheimer's disease (AD) is a persistent disorder of advanced neural activity, referring to a serious primary degenerative brain disease of the central nervous system, which often occurs in the elderly. The course of the disease is slow and irreversible, mainly characterized by progressive cognitive decline. The incidence of AD is rapidly increasing globally (*Martinez & Peplow, 2019*; *Goedert & Spillantini, 2006*; *Scheltens et al., 2021*). The progression of AD has grave adverse effects on the patient's normal life, and brings substantial economic burden to their families and society (*Supekar et al., 2008*; *Lei, Ayton & Bush, 2021*). Until now, people still have no way to cure AD, but it can be stopped or delayed by an early intervention. Thus, it is of great significance to identify the latent phase of AD (mild cognitive impairment, MCI) (*Yu et al., 2017*; *Rostamzadeh et al., 2022*; *Li et al., 2019*), and carry out incipient diagnosis, timely intervention and treatment.

To identify MCI as early as possible, people have come up with lots of approaches to measure the activity and organization of the brain (*Forman et al., 2007*; *Drzezga et al., 2005*). In recent years, resting state functional magnetic resonance imaging (rs-fMRI) (*Huettel, Song & McCarthy, 2004*), which takes the blood oxygen level dependent (BOLD) signals for measuring nerve actions indirectly, provides a non-invasive method for the study of neurological diseases, being generally utilized in medical and neuroscience studies (*Brunetti et al., 2006*; *Canario, Chen & Biswal, 2021*; *Kevin et al., 2008*; *Yang et al., 2021*). The baseline level of BOLD signal varies for each individual. For example, if a person has a faster metabolism, their overall BOLD signal level may be higher than others. In addition, there are differences in breathing and heartbeat among individuals, causing difficulty in distinguishing subjects from normal controls (NC) by comparing the fMRI time series between the subjects (*Fornito, Zalesky & Bullmore, 2016*). Oppositely, BFN based on fMRI can not only quantitatively depict the dependency between BOLD signals, but also provide a reliable medium for exploring the internal mechanisms of the brain (*Poldrack & Farah, 2015*). In practice, researchers generally first estimate BFNs through modelling the interactions between ROIs and use them as features for downstream recognition or classification tasks. Therefore, it is imperative to evaluate BFNs more reasonably.

Because of the importance of BFN, people have put forward a lot of schemes for estimating BFNs in the last few years (*Pervaiz et al., 2020*). In these methods, Pearson's correlation (PC) is renowned for its simplicity and efficiency. Considering that PC always leads to dense BFN, which violates the biological prior of brains. Threshold is often applied to sparse the estimated BFN through cutting off the inferior connections. *Li et al. (2017)* recently invented an $l_1$-norm regularizer for the PC model to get a sparsity BFN (namely $PC_{Sparsity}$). In addition, PC estimates BFNs through measuring the total correlation between different ROIs, but it is inclined to involve hybrid influence from other brain regions. Conversely, partial correlation mitigates the issue through degenerating the latent impact from the other ROIs while computing the edge weights of BFN. Nevertheless, partial correlation calculation concerns with a reverse conducting on the covariance matrix, causing an ill-posed problem at most of the time. Therefore, when estimating

BFNs, people usually introduce an $l_1$-norm regularizer into the partial correlation model, thus bringing about the popular sparse representation (SR) (*Lee et al., 2011*).

Despite their effectiveness, the approaches above did not take the spatial positional relationship between brain regions into account. To overcome the issue, we present a neighborhood structure-guided BFN estimation method. Specifically, based on the obtained brain BOLD signals, we calculate the Euclidean distance between ROIs and sort them. Then, K-nearest neighbor (KNN) algorithm (*Peterson, 2009*) is applied to find the top K neighbors closest to the current ROIs, establish the connection relationship between the ROIs and these K neighbors and construct the global topology adjacency matrix based on the binary network. Based on adjacency matrix, PC calculates the correlation coefficient between ROIs connected by edges, and generates the BFN. Therefore, the proposed method can not only take the brain topology into account, but also generate sparse BFNS, which is in line with the sparsity prior to brain connectivity (*Sporns, 2011*). For evaluating the performance of our presented approach, we employ it to estimate BFNs first. Then we utilize the estimated BFNs to perform MCI classification. The classification results indicate that the presented approach gets better accuracy than all the baselines.

The article is structured as follows: We begin by examining three prominent BFN estimation methods in 'Related Works'. This is followed by an in-depth presentation of our method, delineating the underlying motivation and the model framework, in 'The Proposed Method'. We then detail the data source and experimental settings in 'Experiments'. Subsequent to this, in 'Results', we utilize the presented approach to MCI identification and present our findings. 'Discussions' is dedicated to examining the implications of our results and the factors influencing performance. The article culminates with a conclusion that encapsulates our research.

## RELATED WORKS

On account of the potentiality in studying neurodegenerative diseases of the brain, lots of BFN estimation studies have been reported for the last few years (*Smith et al., 2011*; *Tu & Zhang, 2022*; *Smith et al., 2013*), from the relatively simple and efficient PC to the most complicated dynamic causal modeling (DCM) (*Friston, Harrison & Penny, 2003*). In this section, we briefly look back some approaches, *i.e.,* PC, PC$_{Sparsity}$ and SR, which are closely connected with our work.

### Pearson's correlation

As mentioned earlier, PC is a relatively simple, efficient and popular method for establishing BFNs (*Smith et al., 2013*). We assume that each brain, due to some atlas, is segmented into $n$ ROIs. And the edge weight matrix $W = (w_{ij}) \in R^{n \times n}$ of BFN is given as follows:

$$w_{ij} = \frac{(x_i - \overline{x}_i)^T (x_j - \overline{x}_j)}{\sqrt{(x_i - \overline{x}_i)^T (x_i - \overline{x}_i)} \sqrt{(x_j - \overline{x}_j)^T (x_j - \overline{x}_j)}} \tag{1}$$

where $w_{ij}$ represents the estimated functional connectivity between the $i$th and $j$th ROIs, $x_i \in R^{m \times 1}$ $(i = 1, \ldots, n)$ represents the observed time series (*i.e.,* BOLD signal) in relation
to the $i$th ROIs, $m$ represents the number of the time volumes in every series. All entries in $\bar{x}_i \in R^{m \times 1}$ are the average values of elements in $x_i$, $x_i - \bar{x}_i$ represents the centralization of signal $x_i$. Without loss of generality, we redefine $x_i = (x_i - \bar{x}_i)/\sqrt{(x_i - \bar{x}_i)^T (x_i - \bar{x}_i)}$. Consequently, Eq. (1) can be simplified as $w_{ij} = x_i^T x_j$, or an equivalent matrix form:

$$W = X^T X \tag{2}$$

where $X = [x_1, x_2, \ldots, x_n] \in R^{m \times n}$ represents the rs-fMRI data matrix, $W = (w_{ij}) \in R^{n \times n}$ represents the edge weight matrix of the estimated BFN.

Practically, Eq. (2) can serve as a solution to the following optimization problem (*Li et al., 2017*):

$$\min_W \left\| W - X^T X \right\|_F^2 \tag{3}$$

where $||\cdot||_F$ stands for the Frobenius norm of a matrix.

## Sparse Pearson's correlation

As we all know, because noises are usually included in BOLD signals, the original PC are more prone to bringing about a BFN with intensive links (*Fornito, Zalesky & Bullmore, 2016*). To alleviate this problem, a threshold structure is usually applied to sparse the PC BFNs in practical applications. However, the selection of thresholds is usually empirical and lacks elegant mathematical representations. Therefore, Li et al. introduced the $l_1$-norm as regularization term (*Li et al., 2017*), resulting in the $PC_{\text{Sparsity}}$ model as below:

$$\min_W \left\| W - X^T X \right\|_F^2 + \lambda ||W||_1 \tag{4}$$

where $\lambda$ represents a regularized parameter to controll the sparsity of $W$. Apparently, the $PC_{\text{Sparsity}}$ decreases to the original PC as $\lambda = 0$. Here, we select the $l_1$-norm to sparse the network since it is simple and popular.

## Sparse representation

Unlike PC-based methodologies that measure the total correlation between ROIs, through degenerating the hybrid influence of other ROIs, partial correlations may cause more correlated BFN estimation, which depicts a relatively reliable interaction between brain regions (*Peng et al., 2009*). A universal method for calculating partial correlation is founded on the inverse covariance matrix estimation. Nevertheless, because of the singularity of the covariance matrix, the estimation of partial correlation may be maladaptive. For acquiring a steady estimate of BFN, an $l_1$-norm regularization term is usually presented in the partial correlation model. For instance, partial correlation estimation based on SR is a classic statistical method. The mathematical model of SR is represented as below:

$$\min_{w_{ij}} \sum_{i=1}^{n} \left( \left\| x_i - \sum_{j \neq i} w_{ij} x_j \right\|^2 + \lambda \sum_{j \neq i} \left| w_{ij} \right| \right)$$

$$s.t. w_{ii} = 0, \forall i = 1, \ldots, n \tag{5}$$

 

Then, Eq. (5) can be further simplified to:

$$\min_{W} \|X - XW\|_F^2 + \lambda \|W\|_1$$

$$s.t. \, w_{ii} = 0, \forall i = 1, \dots, n \qquad (6)$$

where $\|\cdot\|_F$ and $\|\cdot\|_1$ represent the $F$-norm and $l_1$- norm of a matrix. $\|X - XW\|_F^2$ represents a adapting term, $\|W\|_1$ represents a $l_1$-regularized term used to get sparse solutions of $W$, $\lambda$ represents the regularized parameter to control the equilibrium of two terms in Eq. (6). The constraint $w_{ii} = 0$ is applied for removing $x_i$ from $X$ to stay away from the problem of trivial solution.

## THE PROPOSED METHOD

### Motivation

As described above, the classification performance of the brain disease recognition task critically depends on the quality of the constructed BFNs. Therefore, many BFNs estimation schemes have been proposed over the last few decades, and the three methods (*i.e.*, PC, $PC_{Sparsity}$ and SR) introduced in the previous section are the most common statistical methods. These three methods, whether using a hard-threshold to remove weak connection edges or introducing $l_1$-norm as a regularization term to generate an elegant mathematical representation to build the network, can ultimately obtain sparse BFNs, which is consistent with the scant prior information on brain network (*Sporns, 2011*). Although the above methods have been widely used in neurological diseases, the topological structure of brain networks is ignored in estimating BFNs. Besides, recent studies have indicated that due to the principle of simplicity in wiring costs, ROI closer in space has higher connectivity prospects than ROI farther in space (*Fornito, Zalesky & Bullmore, 2016*). Due to all the viewpoints above, we present a new BFN estimation method, which uses spatial distance information to reconstruct the PC model. The details of our method will be introduced in the following section.

### Model and algorithm

In order to obtain sparse BFNs and incorporate the topology structure information, we propose a neighborhood structure-guided BFN construction scheme in this article. The whole framework is divided into the following steps.

*Step1:* Calculate the Euclidean distance between ROIs. There are many methods to measure distance, including Euclidean distance, Cosine similarity, Manhattan distance, and so on. Cosine similarity is more suitable for measuring the directionality and correlation between vectors, without considering their absolute value size. The Manhattan distance, on the other hand, is not as direct when considering the distances of various dimensions as the Euclidean distance, in some cases, it may not be possible to capture the feature relationships of the data. So, we choose the most suitable and commonly used Euclidean
distance as the measurement strategy of spatial distance information between ROIs. The mathematical calculation formula for Euclidean distance is as follows:

$$d_{ij} = \sqrt{\left(x_{1,i} - x_{1,j}\right)^2 + \left(x_{2,i} - x_{2,j}\right)^2 + \cdots \left(x_{m,i} - x_{m,j}\right)^2}$$

$$= \sqrt{\sum_{l=1}^{m}\left(x_{l,i} - x_{l,j}\right)^2} \tag{7}$$

where $x_i = (x_{1,i}, x_{2,i}, \ldots, x_{m,i})^T$ represents the mean time series extracted from the $i$th ROIs, $d_{ij}$ represents the distance between the $i$th and $j$th ROIs. The distance matrix $D = (d_{ij}) \in R^{n \times n}$ is obtained by calculating the distance between all ROIs using Eq. (7).

*Step2*: Construct the adjacency matrix with topological structure. The distance between each ROIs and other ROIs is sorted from smallest to most significant according to the distance matrix $D$. Then, the KNN algorithm is used to find the $k$ ROIs closest to the current ROIs (that is, the top $k$ ROIs with the smallest distance) as neighbors and connect them with edges to generate an adjacency graph. The details are shown in Fig. 1. This graph can be represented by the following adjacency matrix $S = (s_{ij}) \in R^{n \times n}$:

$$s_{ij} = \begin{cases} 1, & e_{ij} \text{ exists} \\ 0, & \text{otherwise} \end{cases} \tag{8}$$

where $e_{ij}$ is the edge between the $i$th and $j$th ROIs. Eq. (8) indicates that if an edge is connected between the $i$th and $j$th ROIs, the edge weight $s_{ij}$ is regarded as 1. Otherwise, the edge weight $s_{ij}$ is 0. In this way, the adjacency matrix $S$ that only distinguishes between connected and unconnected networks is obtained. This is in line with the previous introduction that ROIs with close spatial distance are more inclined to be connected by edges. The selection of $k$ value in KNN algorithm will be discussed in 'Discussions'.

*Step3*: Estimating BFNs based on adjacency matrix $S$. According to matrix $S$, we propose a new sparse PC model for estimating BFNs. Specifically, the calculation formula of this model is as follows:

$$w_{ij} = \begin{cases} ll \dfrac{(x_i - \overline{x}_i)^T (x_j - \overline{x}_j)}{\sqrt{(x_i - \overline{x}_i)^T (x_i - \overline{x}_i)}\sqrt{(x_j - \overline{x}_j)^T (x_j - \overline{x}_j)}}, & \text{if } s_{ij} = 1 \\ 0, & \text{otherwise} \end{cases} \tag{9}$$

where $w_{ij}$ represents the edge weight between the $i$th and $j$th ROIs. According to Eq. (9), the correlation coefficients between all ROIs with edges connected can be calculated as the edge weights $w_{ij}$. As a result, the sparse BFN containing topology can be finally generated. For facilitating the subsequent description, the BFN constructed by the KNN is abbreviated as K-BFN.

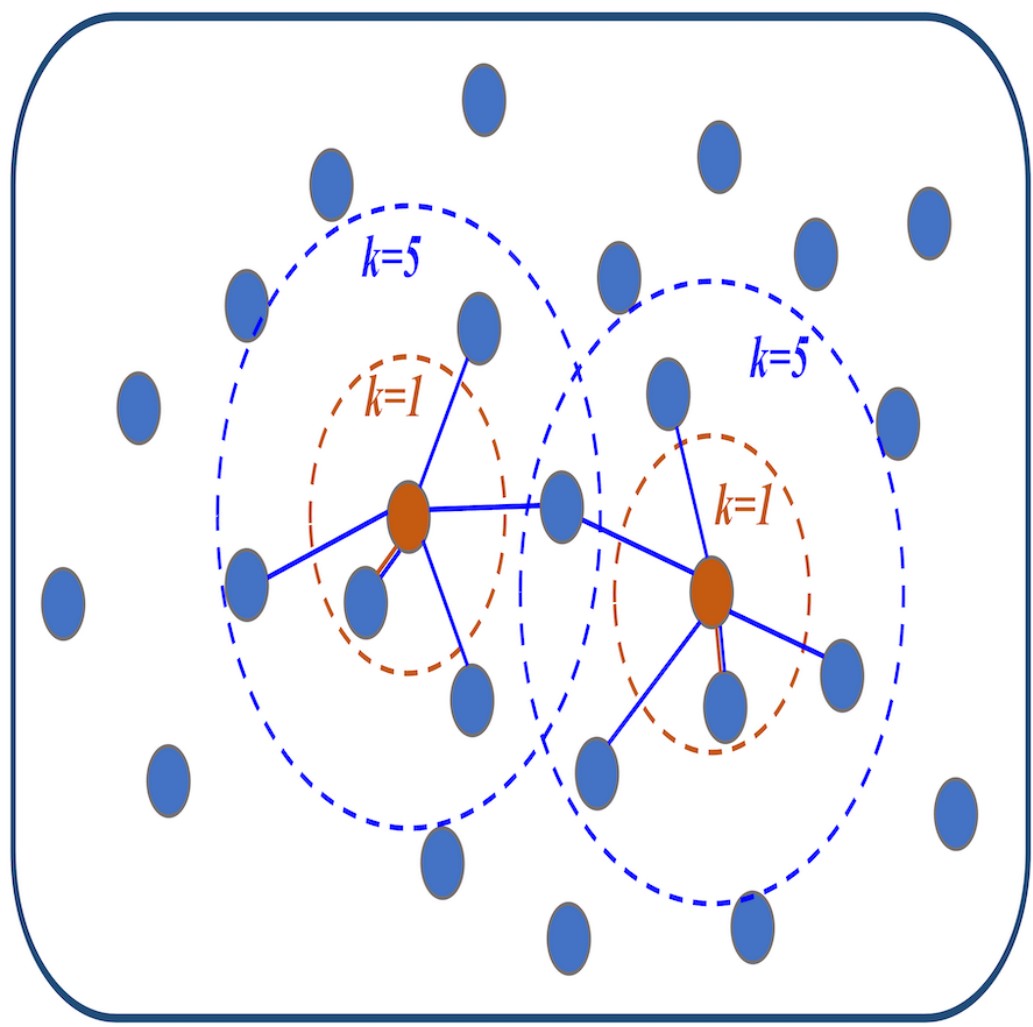

**Figure 1** The visualization diagram of the K-nearest neighbor (KNN) algorithm.

## EXPERIMENTS

### Data acquisition and preprocessing

For evaluating the performance of the presented approach, we conduct MCI recognition task in accordance with the benchmark database, *i.e.,* Alzheimer's Disease Neuroimaging Initiative (ADNI) dataset in the experiment. Concretely, 137 subjects (68 MCIs, 69 NCs) were participated as in the researches (*Zhou et al., 2018*; *Van der Haar et al., 2023*; *Sun et al., 2021*). All these data are publicly available in the ADNI database (http://adni.loni.usc.edu/). All subjects were scanned through a 3.0T Philips scanner, with the scanning time of 7 min, equivalent to 140 volumes. Table 1 shows the demographic information of these subjects.

For decreasing the impact of interference signals, the preprocessing pipeline in SPM8 toolbox (http://www.fil.ion.ucl.ac.uk/spm/) was applied to enhance the quality of fMRI data before BFN estimation. Especially, the first 3 volumes of one subject in the fMRI time course were removed to keep the signal stability. Next, the volumes leave behind were

**Table 1** **Demographic and clinical information of subjects in the ADNI datasets.** Values are reported as mean ± standardard deviation. M/F: Male/Female.

| Dataset | Class | Age (Years) | Gender(M/F) |
|---------|-------|-------------|-------------|
| ADNI | MCI | $76.50 \pm 13.50$ | 39/29 |
| | NC | $71.50 \pm 14.50$ | 17/52 |

handled based on a popularly-used framework. The major stages were (1) modifying slice timing and head movement (*Friston et al., 1996*); (2) fitting the ventricle and white matter signals with Friston 24-parameters of head movement; (3) enrolling the modified rs-fMRI graphics to Montreal Neurological Institute (MNI) standard space (*Brett et al., 2001*); (4) performing spatial smoothing on the signal with a maximum half width of four mm and using a bandpass filter (0.015–0.15 Hz) on it. In the end, each brain was divided into 116 ROIs due to the Automated Anatomical Labeling (AAL) template (*Tzourio-Mazoyer et al., 2002*). The average time series of each ROI for each subject were extracted and placed in the data matrix $X \in R^{137 \times 116}$ which is just used to estimate BFNs.

## Experimental setting
### Brain functional network estimation
The pre-processed time series form a data matrix for each subject to estimate the BFN. In our experiments, we first compare the proposed approach K-PC with three typical baselines for BFN construction, *i.e.,* PC, PC$_{\text{Sparsity}}$ and SR. And each BFN construction baseline involves a free parameter. Due to the significant impact of modeling parameters on the structure and classification performance of the network model, the optimal parameters were selected through large-scale grid search. Specially, in terms of the regularized parameters(*i.e.,* $\lambda$ in PC$_{\text{Sparsity}}$, $\lambda$ in SR), the range of candidate values is $[2^{-5}, 2^{-4}, \ldots, 2^4, 2^5]$. In specific, the presented approach K-PC has a hidden parameter $k$, which controls the number of neighbors in the KNN algorithm when estimating the BFN. Firstly, we appoint a value to $k$ by experience. Secondly, we study the significance of $k$ in the range of $[110, 100, \ldots, 20, 10]$ in 'Discussions'. In regard to PC, the model itself does not involve free parameters. Towards a reasonable comparison with the baselines and modification of its flexibility, we present a threshold parameter into PC to cut off the weakly connected edges in the estimated BFNs. For the sake of fairness, we use 11 thresholds corresponding to varying degrees of sparsity $[0, 10\%, \ldots, 90\%, 99\%]$, where the percentage represents the percentage of weak connections removed.

### Feature selection and classification
After gaining the BFNs, the follow-up mission is identifying MCI and NC due to the estimated BFNs. Now, the issue becomes the decision of which features and classifiers are going to be selected for MCI recognition. Given that feature selection and classifier design have significant effects on the identification performance, it's ambiguous to tell whether the BFN estimation approaches or the follow-up steps in the follow-up pipeline is beneficial for the outcome. Hence, we merely take a relatively simple and efficient feature selection way ($t$-test with four accepted $p$-values) and a very popular used support vector machine (SVM) classifier (*Chang & Lin, 2011*), as our primary focus is BFN estimation.

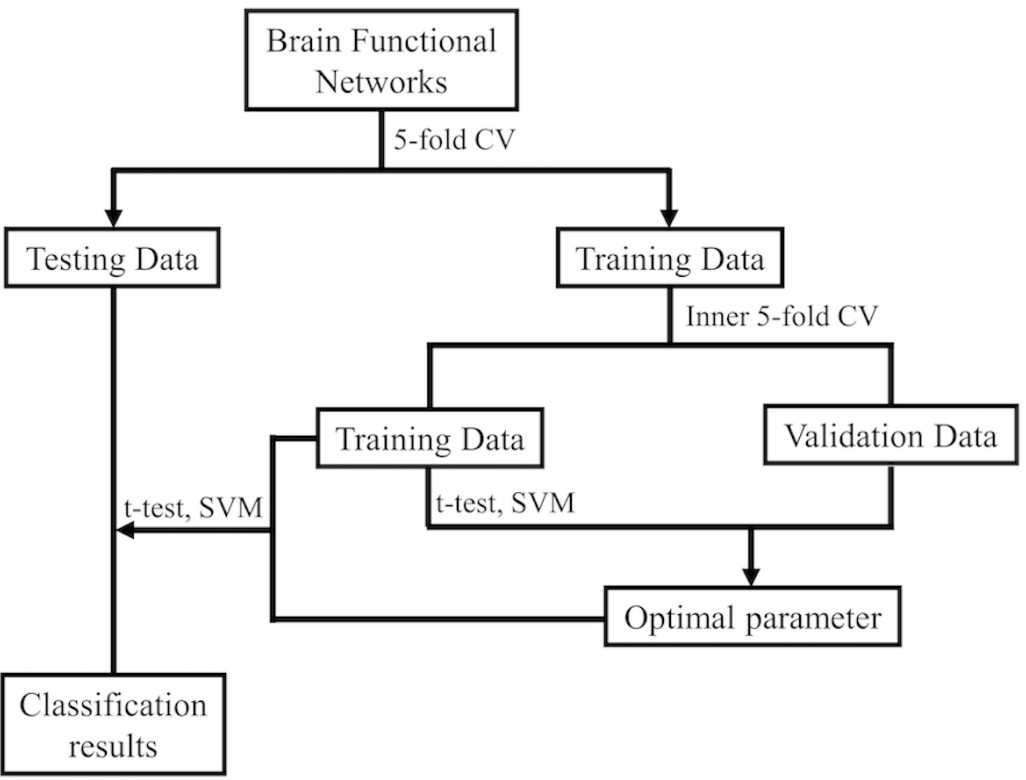

**Figure 2   The MCI identification pipeline based on the estimated BFNs.**

Figure 2 shows the basic procedure to classify MCI and NC using our designed pipeline. In the experiment, weights between ROIs are regarded as features for identification. As the AAL atlas contains 116 ROIs, there will be $116 \times (116 - 1)/2 = 6670$ connections/attributes for each BFN, which is far greater than the sample size (the number of subjects applied in the experiment), resulting in the collapse of dimensionality. For overcoming the issue, we apply two-sample $t$-test for feature selection in advance of MCI identifications.

Further, to make full use of the limited samples, *i.e.,* the estimated BFNs of distinct subjects, during the training phase, we use 5-fold cross validation 100 times to investigate the effectiveness of the relevant approaches (*Qiu et al., 2020*; *Diogo, Ferreira & Prata, 2022*). Specifically, the 137 subjects of ADNI are first divided into five folds, each containing nearly the equivalent amount of connections the samples. Then, four folds are applied for training, and the other one is for testing. Note that the same linear SVM with default parameter $C = 1$ is applied in every method used in this work for fair. As shown in Fig. 2, an inner 5-fold CV is operated through grid-search to choose the optimal parameters. The accuracy is computed by averaging the outcomes of all subjects.
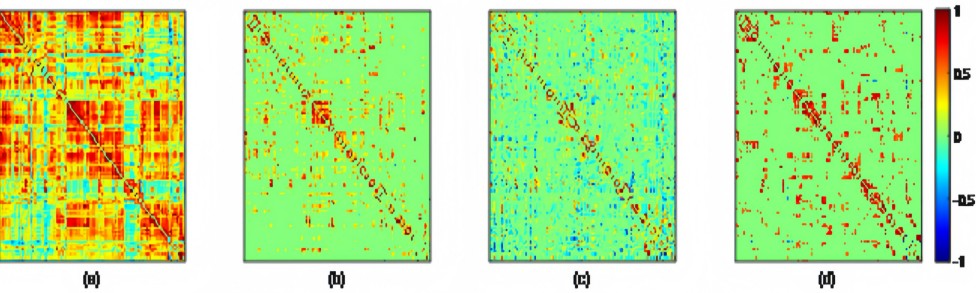

**Figure 3** The adjacency matrices of the same subject estimated by four different methods, *i.e.*, (A) PC, (B) PC$_{Sparsity}$, (C) SR, and (D) K-PC.

## RESULTS

### Network visualization

Using the preprocessed rs-fMRI data, we get the modified BFNs with the presented K-PC methods. Moreover, we construct BFNs for comparison according to the baselines including PC, PC$_{Sparsity}$ and SR. Figure 3 shows the adjacency matrices of BFNs constructed by four various approaches for MCI recognition (the $K = 40$). From Fig. 3, we can observe that BFNs estimated based on PC$_{Sparsity}$, SR and K-PC methods are very clear compared with PC. The adjacency matrices of PC based BFNs are too divergent, and compared to them, the adjacency matrices of the other three methods are more clustered in a certain type of linear regression. And the BFN is not sparse, as pairwise complete correlation is applied for modelling the network adjacency matrix. Consequently, it results in new incorrect connections, affecting the final classification accuracy. Conversely, the networks on the basis of PC$_{Sparsity}$ and SR are sparse because of the introduction of the $l_1$-norm regularizer. Besides, it is worth noting that the BFN estimated through K-PC exhibits a more pronounced module topological architecture than other methods. To evaluate the modularity of BFNs, the signed modularity maximization method is applied for calculating the modularity scores of BFNs constructed through various approaches (*Gómez, Jensen & Arenas, 2009*; *Rubinov & Sporns, 2011*). Moreover, for keeping off the randomness of the results, 28 subjects are selected equidistantly. As Fig. 4 shown, the BFN estimated by K-PC is significantly more modular than the other three baseline methods, indicating that K-PC tends to achieve cleaner networks. Additionally, modularization of BFN estimated by PC$_{Sparsity}$, SR and K-PC acquire higher scores than that by PC in most cases, suggesting that moderate sparsity promotes the modularity architecture of the network.

### Classification performance

An array of quantitative mensuration is conducted to test the performance of different approaches, like classification accuracy (ACC), sensitivity (SEN), specificity (SPE), positive predictive value (PPV) and negative predictive value (NPV). The definitions of each method are shown in Table 2, where TP, TN, FP, and FN represent true positive, true negative, false positive and false negative.

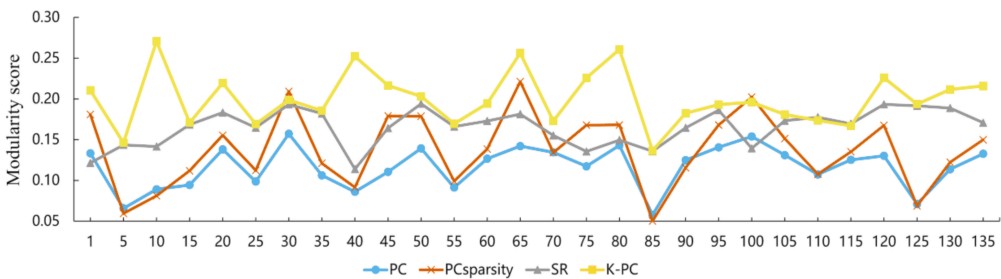

**Figure 4** **Modularity scores of BFN constructed by different methods.** Note that the abscissa indicates the label of the participant.

**Table 2** **Definitions of the performance metrics involved in this study.**

| Performance metric | Abbreviation | Definition |
|---|---|---|
| Accuracy | ACC | $\frac{TP+TN}{TP+TN+FP+FN}$ |
| Sensitivity | SEN | $\frac{TP}{TP+FN}$ |
| Specificity | SPE | $\frac{TN}{TN+FP}$ |
| Positive Predictive Value | PPV | $\frac{TP}{TP+FP}$ |
| Negative Predictive Value | NPV | $\frac{TN}{TN+FN}$ |

Table 3 reports the MCI recognition outcomes of different approaches under five evaluation metrics. As shown in Table 3, our presented K-PC exceeds the baselines according to the average ACC score under the circumstance that $p$-values are 0.01, 0.05 and 0.001 in accordance with 5-fold CV. Note that our K-PC gets the ACC of 81.75%, SEN of 83.09%, SPE of 80.43%, PPV of 80.70% and NPV of 82.87% when $p$-value is 0.05, which are significantly better than the baselines. Regarding the situation where the $p$-value is 0.005, our K-PC, *i.e.,* even if the ACC, SEN and NPV performances are not satisfactory, also gets prominently better results than the other compared approaches on SPE and PPV. The outcome of the experiments evidence that the presented K-PC framework can enhance the baselines, *i.e.,* at least the discrimination of the estimated BFNs.

For verifying the validity of the presented approach, we compared it with the most commonly used time series methods in deep learning. We chose classic BiGRU, BiLSTM, and Transformer for comparison. From Table 4, it can be seen that among these methods, Transformer performs the best in terms of five performance indicators. The time series methods such as Transformer typically directly model raw data without an accurate feature selection process. They rely on the learning ability of the model to learn the most relevant features from the raw data. When selecting the appropriate $p$-value, the K-PC method outperforms the traditional time series prediction methods. The common algorithms for deep learning exhibit unstable stability, with significant performance differences between each fold, indicating poor generalization ability and excellent performance in capturing complex relationships in MCI data. Among the five evaluation metrics (*i.e.,* ACC, SEN, SPE, PPV and NPV), the degree of variation in the K-PC method was 1 / 7 to 1 / 5 that of

**Table 3** **The classification performance (mean ± standard deviation) of different BFN estimation methods based on five performance metrics (*i.e.*, ACC, SEN, SPE, PPV and NPV) using 5-fold CV.**

| *p*-value | Method | ACC | SEN | SPE | PPV | NPV |
|---|---|---|---|---|---|---|
| $p = 0.01$ | PC | 0.6905 ± 0.0052 | 0.7015 ± 0.0312 | 0.6797 ± 0.0205 | 0.6834 ± 0.0064 | 0.6991 ± 0.0179 |
| | PC$_{Sparsity}$ | 0.7394 ± 0.0206 | 0.7000 ± 0.0104 | 0.7783 ± 0.0307 | 0.7588 ± 0.0255 | 0.7241 ± 0.0151 |
| | SR | 0.7336 ± 0.0052 | 0.7426 ± 0.0312 | 0.7246 ± 0.0205 | 0.7268 ± 0.0064 | 0.7412 ± 0.0179 |
| | K-PC | **0.7993 ± 0.0103** | **0.8015 ± 0.0104** | **0.7971 ± 0.0102** | **0.7964 ± 0.0112** | **0.8029 ± 0.0096** |
| $p = 0.05$ | PC | 0.7664 ± 0.0155 | 0.7721 ± 0.0104 | 0.7609 ± 0.0410 | 0.7607 ± 0.0312 | 0.7723 ± 0.0016 |
| | PC$_{Sparsity}$ | 0.7810 ± 0.0118 | 0.7676 ± 0.0197 | 0.7942 ± 0.0171 | 0.7868 ± 0.0125 | 0.7768 ± 0.0147 |
| | SR | 0.7591 ± 0.0103 | 0.7868 ± 0.0312 | 0.7319 ± 0.0102 | 0.7426 ± 0.0029 | 0.7794 ± 0.0251 |
| | K-PC | **0.8175 ± 0.0155** | **0.8309 ± 0.0104** | **0.8043 ± 0.0205** | **0.8070 ± 0.0191** | **0.8287 ± 0.0123** |
| p=0.001 | PC | 0.6496 ± 0.0103 | 0.6103 ± 0.0312 | 0.6884 ± 0.0102 | 0.6586 ± 0.0041 | 0.6422 ± 0.0150 |
| | PC$_{Sparsity}$ | 0.6701 ± 0.0166 | 0.6206 ± 0.0351 | 0.7188 ± 0.0079 | 0.6855 ± 0.0111 | 0.6579 ± 0.0242 |
| | SR | 0.6971 ± 0.0258 | 0.7353 ± 0.0624 | 0.6594 ± 0.0102 | 0.6799 ± 0.0119 | 0.7183 ± 0.0449 |
| | K-PC | **0.7445 ± 0.0103** | **0.7574 ± 0.0208** | **0.7319 ± 0.0410** | **0.7357 ± 0.0235** | **0.7537 ± 0.0003** |
| $p = 0.005$ | PC | 0.6241 ± 0.0258 | 0.6471 ± 0.0416 | 0.6014 ± 0.0102 | 0.6151 ± 0.0213 | 0.6341 ± 0.0313 |
| | PC$_{Sparsity}$ | 0.7095 ± 0.0140 | 0.7206 ± 0.0197 | 0.6986 ± 0.0130 | 0.7021 ± 0.0125 | 0.7179 ± 0.0169 |
| | SR | **0.7226 ± 0.0052** | **0.7279 ± 0.0104** | 0.7174 ± 0.0205 | 0.7191 ± 0.0113 | **0.7271 ± 0.0045** |
| | K-PC | 0.7117 ± 0.0052 | 0.6765 ± 0.0208 | **0.7464 ± 0.0102** | **0.7240 ± 0.0024** | 0.7019 ± 0.0154 |

**Notes.**
Note that the bold value indicates the best result.

**Table 4** **The classification performance (mean ± standard deviation) of our method and traditional time series classification method based on five performance metrics using 5-fold CV result.**

| Method | ACC | SEN | SPE | PPV | NPV |
|---|---|---|---|---|---|
| BiGRU | 0.6638 ± 0.0728 | 0.7494 ± 0.0835 | 0.5802 ± 0.1253 | 0.6411 ± 0.0699 | 0.7011 ± 0.0769 |
| BiLSTM | 0.7008 ± 0.0293 | 0.7241 ± 0.0882 | 0.6659 ± 0.0847 | 0.6873 ± 0.0411 | 0.7263 ± 0.0526 |
| Transformer | **0.7659 ± 0.0867** | **0.7791 ± 0.0748** | **0.7538 ± 0.1188** | **0.7623 ± 0.1008** | **0.7743 ± 0.0809** |

**Notes.**
Note that the bold value indicates the best result.

other traditional time-series approaches. In the field of MCI identification outcome, we can consider the K-PC method superior to deep learning traditional temporal methods.

## DISCUSSIONS

In this section, we study the influence of feature selection and network modeling parameters on the identification outcome (*Jiang et al., 2019*; *Zhang et al., 2022*; *Xue et al., 2020*; *Jiao et al., 2022*). Then, we discuss the most representative features obtained through the proposed approach to study the relationship with brain diseases.

### Parameter analysis

It is well known that feature selection is a widely used dimensionality reduction technique, which can effectively reduce redundant features in data. Therefore, it is also generally applied in the research of neurological diseases as per fMRI data. Now, there are many proposed feature selection approaches. In this article, we only focus on two-sample *t*-test, since it is relatively simple and efficient in practice. Specifically, we conducted comparative

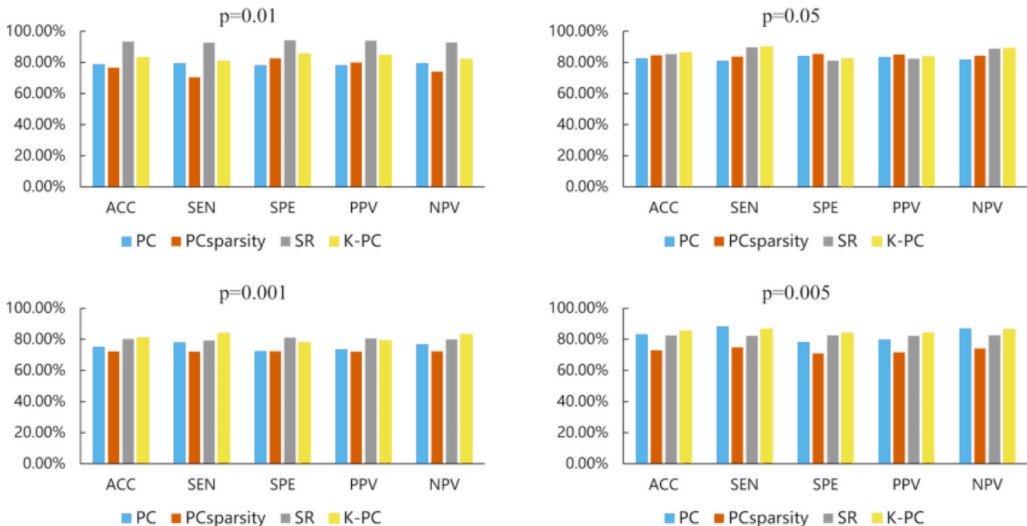

**Figure 5** **The MCI classification results based on four methods (*i.e.*, PC, PC$_{Sparsity}$, SR and K-PC) under optimal parameter values. In particular, each subgraph represents the classification results under five performance metrics based on different p.** In particular, each subgraph represents the classification results under five performance metrics based on different *p*-values (0.01, 0.05, 0.001, 0.005) involved in the *t*-test.

analysis based on the *t*-test with four various *p*-values, *i.e.,* 0.01, 0.05, 0.001, 0.005, $k = 40$, in the experiment, and then show the classification results in Fig. 5. Compared with the outcomes, we can see the following facts: when using *t*-test under the $p = 0.05$, our method can obtain the best classification accuracy, of 86.13%. However, we note that compared with other methods, the SR-based method can achieve a better final accuracy of 93.43% with *p*-values of 0.01, and even the classification performance is much higher than our K-PC. The phenomenon suggests that the selection of the *P* value is crucial to the classification outcomes, so in the following discussion, we choose $p = 0.05$ as the parameter of *t*-test for feature selection. In conclusion, appropriate *p*-value selection has a positive impact on achieving better identification outcomes.

## Sensitivity to network modelling parameters

Virtually, in addition to the *p*-values applied in the feature selection stage, the network modeling parameters usually have an essential impact on the structure of BFN and the subsequent classification accuracy. For studying the sensitivity of distinct approaches to relevant parameters, we calculate the evaluation metric with various parameters (we set $p = 0.05$ in this part for better performance according to the result from 6.1).

As the experimental outcomes shown in Fig. 6, the variation in classification accuracy corresponds to that in parametric values, suggesting that the majority of the BFN estimation approaches are easily affected by the free parameters. In terms of PC, it can be found that moderate sparsity inclines to lead to higher classification accuracies. Interestingly, the dense BFNs without thresholding operation (parameter value = 0) can achieve high accuracy as well, which may be due to the use of a *t*-test to remove weak discriminative

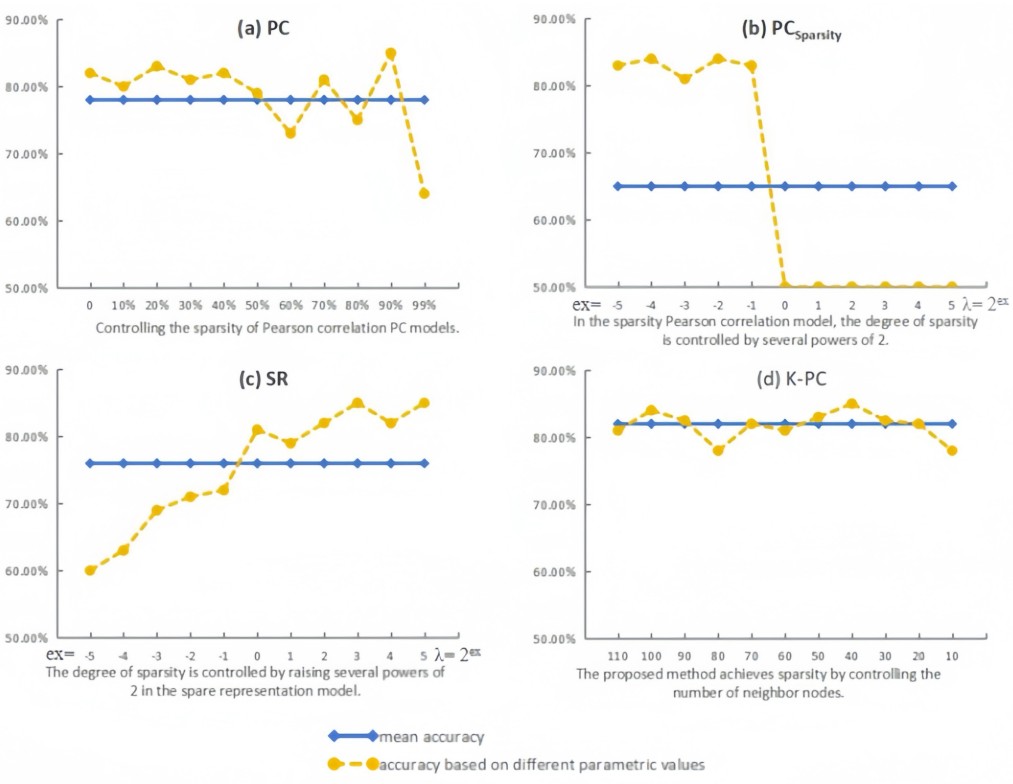

**Figure 6** Classification accuracy by PC, PC$_{Sparsity}$, SR and the proposed method K-PC based on different parametric values.

connections before classification. Compared with PC, PC$_{Sparsity}$ can obtain relatively high recognition results of brain diseases in the parameter interval $[2^{-5}, 2^{-1}]$, indicating that the sparsified BFNs can effectively remove the redundant connections in PC, thus improving the recognition accuracy of patients with brain diseases. However, when the parameter is $2^0$, the accuracy of PC$_{Sparsity}$ is reduced to 50.36%, and the accuracy of PC$_{Sparsity}$ does not change when the parameter is more than $2^0$, which proves that being too sparse can actually lead to a decrease in classification accuracy. For SR, $l_1$-regularizer in the BFN estimation model is applied to remove the weak connections and control the estimated BFN density. From Fig. 6C, it can be found that the best value of $\lambda$ is $2^3$ for SR, and the corresponding accuracy is 85.43%. In particular, according to Fig. 6D, we can find that although the classification performance of K-PC fluctuates with the number of neighbors k, it is least affected by the parameters compared with the three baseline methods. More importantly, K-PC obtains no less than 78% results in the entire parameter range $[110, 100, \ldots, 20, 10]$, which indicates that its recognition performance is more advantageous relative to other methods.

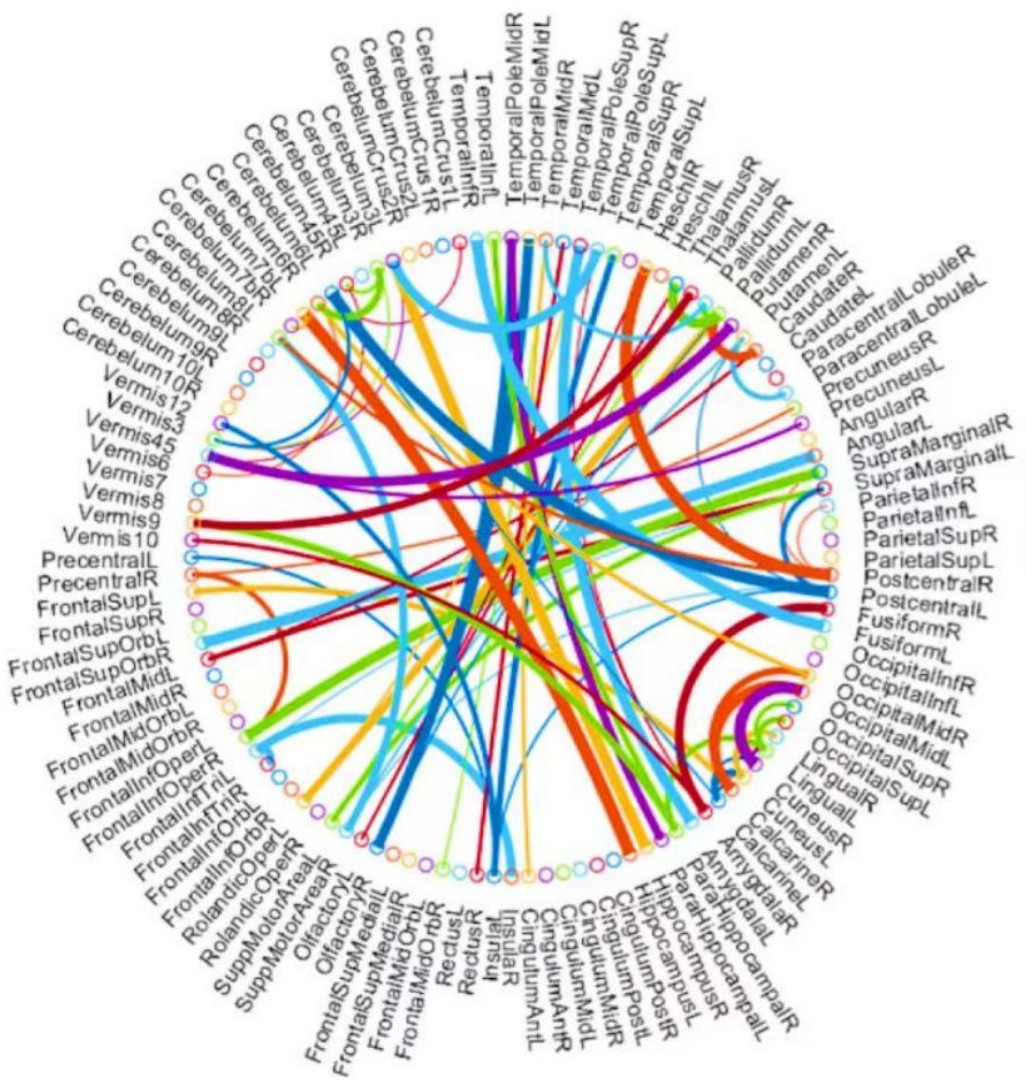

**Figure 7** The most discriminative features (network connections) involved in the classification tasks by using a *t*-test with *p* < 0.005.

## Top discriminative features

Beyond the evaluation metrics, it is more enlightening to find which features/connections in BFN will result in better recognition performance. For BFNs based on K-PC, we study some favorable clues to recognize subjects with MCI from NCs. Particularly, we choose 80 connections whose *p*-values are lower than 0.005. As can be seen from Fig. 7, the gauge of the arc is in reverse proportion to the *p*-value, suggesting the discriminative ability of the corresponding edge. In terms of MCI identification, we sort the brain regions according to the chosen discriminative features and identify the first three, hippocampus, left cerebellum 6, and amygdala. They are generally considered to have biological significance for MCI identification, according to the researches (*Greicius, 2008*; *Albert et al., 2011*) for the last

few years, which further validates the effectiveness of our K-PC method in constructing reliable functional brain networks for MCI identification.

However, this method also has its limitations because the *p*-value indicates the degree of influence of this feature. If the *p*-value is smaller, the significance of the feature's influence is higher. Therefore, features with a *p*-value less than 0.005 are selected. However, the smaller the threshold value, the fewer features will be selected, which may impose certain restrictions on forming a more accurate, clearer, and continuous boundary highlighting MCI features.

From the perspective of algorithm optimization, deep neural network (DNN) offers notable advantages over KNN in terms of feature learning, handling high-dimensional data, generalization ability, and computational efficiency. However, in the context of this study involving small-scale datasets and interpretability requirements, KNN remains a viable option. Nevertheless, further optimization is necessary to strike a balance in defining feature boundaries.

## CONCLUSION

In this work, we present a neighborhood structure-guided BFN construction approach, K-PC, which considers the brain topology and can achieve sparsity, making the estimated BFN more discriminant. To prove the validity of K-PC, we utilize our model to recognize subjects with MCI from NCs. Furthermore, we select PC, $PC_{Sparsity}$ and SR as the baselines, and the classification performance shows that K-PC outperforms the baseline methods. Finally, it is worthwhile mentioning that although the proposed method is helpful to the final classification performance, it involves the problem of artificially selecting the optimal parameters. Therefore, from now on, we are going to exploit BFN algorithms that can automatically learn parameters to obtain more discriminative features, thus improving the final classification accuracy. When compared to three representative traditional time series classification methods (Transformer, BiLSTM, and BiGRU), the results indicate that the K-PC method outperforms traditional time series classification methods and exhibits greater robustness and stability.

### Funding

This work was supported by the China Department of Science and Technology under Key Grant 2023YFE0204300, by the R&D project of Pazhou Lab (HuangPu) under Grant 2023K0606, by the NSFC under Grant 62371476, Grant 12126610, Grant 82371917, Grant 81971691, Grant 81801809, Grant 81830052, Grant 81827802, and Grant U1811461, by the China Department of Science and Technology under Key Grant 210YBXM 2020109002, by the Guangzhou Science and Technology bureau under Grant 2023B03J1237, by the Science and Technology Innovative Project of Guangdong Province under Grant 2018B030312002, by Guangdong Province Key Laboratory of Computational Science at the Sun Yat-sen University under grant 2020B1212060032, by Key-Area Research and Development

Program of Guangdong Province under Grant 2021B0101190003. There was no additional external funding received for this study. The funders had no role in study design, data collection and analysis, decision to publish, or preparation of the manuscript.

## Grant Disclosures

The following grant information was disclosed by the authors:

China Department of Science and Technology: 2023YFE0204300.

R&D project of Pazhou Lab (HuangPu): 2023K0606.

NSFC: 62371476, 12126610, 82371917, 81971691, 81801809, 81830052, 81827802, U1811461.

The China Department of Science and Technology: 210YBXM 2020109002.

The Guangzhou Science and Technology: 2023B03J1237.

The Guangzhou Science and Technology: 2023B03J1237.

The Science and Technology Innovative Project of Guangdong Province: 2018B030312002.

Guangdong Province Key Laboratory of Computational Science at the Sun Yat-sen University: 2020B1212060032.

Key-Area Research and Development Program of Guangdong Province: 2021B0101190003.

## Competing Interests

The authors declare there are no competing interests.

## Author Contributions

- Lizhong Liang performed the experiments, analyzed the data, authored or reviewed drafts of the article, and approved the final draft.
- Zijian Zhu analyzed the data, authored or reviewed drafts of the article, and approved the final draft.
- Hui Su analyzed the data, prepared figures and/or tables, and approved the final draft.
- Tianming Zhao analyzed the data, authored or reviewed drafts of the article, and approved the final draft.
- Yao Lu conceived and designed the experiments, analyzed the data, authored or reviewed drafts of the article, and approved the final draft.

## Data Availability

The original data are available in the Supplementary Files.

All the data used in the experiments, as well as the models generated and the source code are available at GitHub and Zenodo:

-https://github.com/zhaotm/NSGBFN.

- Liang, L. (2024). Neighborhood Structure-Guided Brain Functional Networks Estimation for MCI Identification. Zenodo. https://doi.org/10.5281/zenodo.10903092.

## Supplemental Information

Supplemental information for this article can be found online at http://dx.doi.org/10.7717/peerj.17774#supplemental-information.

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
