# Peer review of "Neighborhood structure-guided brain functional networks estimation for mild cognitive impairment identification"

_PeerJ, doi:10.7717/peerj.17774_

## Round 0.1 · original submission · Major Revisions

The authors should carefully address the concerns.

·

Basic reporting

This paper proposes a neighborhood structure-guided brain functional networks estimation method and validates its reliability on MCI identification. Overall, the paper is well-argued and clearly articulated. However, the following questions need to be addressed.
1.What is the ‘biological prior’ mentioned in lines 25 and 72? Does it refer to ‘sparsity’? We suggest that the authors provide literatures or evidences supporting the sparsity of brain networks.
2.Consider whether the term ‘PC-based’ in line 26 should be removed for a better readability.
3.Please explain the sentence that ‘We establish the connection relationship between the ROIs and these K neighbors and construct the global topology adjacency matrix according to the binary network’ in line 31.
4.Clarify whether ‘the adjacency matrix’ mentioned in line 33 refers to the previously described binary matrix. Additionally, clarity is needed for other mentions of the adjacency and binary matrices.
5.Please provide a definition for ‘k-PC’ as mentioned in line 38, and then revise the sentence that ‘the BFN constructed by the KNN is abbreviated as K-PC’ in line 208 according to the definition.
6.Ensure consistency in the capitalization of the word ‘correlation’ in lines 70 and 23.
7.In line 124, the usage of the word ‘resolution’ is not correct.
8.It may strengthen the paper if the information from line 169 about the principle of simplicity in wiring costs and ROI proximity on connectivity is also included in the abstract and introduction as evidence supporting the proposed neighborhood structure-guided BFN construction.

Experimental design

Overall, the description of the experimental design is clear. However, there is some questions need to be addressed.
1.In experiments, why not consider testing other distance metrics such as Cosine similarity or Manhattan distance when constructing the k neighborhood to bolster the persuasiveness of the proposed method.
2. Please correct ‘exist’ to ‘exists’ in line 193.
3. In line 259, it is better to replace the parentheses with ‘i.e.,’. Similarly, in lines 297, 300 and 326, the usage of the parentheses is problematic.
4. In line 260, the ‘5-fold’ in the parenthese should be deleted.
5. In line 262, what is the meaning of ‘the issues’?

Validity of the findings

1. Describe the criteria for selecting the subject whose adjacency matrices are displayed in Figure 3.
2. Please explain the statement that ‘SR and K-PC methods are very clear compared with PC’ in line 273.
3. Figure 5 shows high classification accuracies of 86.13% and 93.43%, which, to our best knowledge, are rare for the ADNI dataset. We suggest the authors provide a further explanation about this point, or share the code for the purpose of easy reproducibility of experimental results.
4. What does the x-coordinate represent in Figure 6? Please supplement this information in the figure itself and the corresponding text description.
5. Why are there no corresponding results for the proposed method in Table 4?

Reviewer 2 ·

Basic reporting

In this paper, a new BFN construction method is proposed, which takes into account the brain's neighborhood structure. The effectiveness of the method was validated by identifying subjects with mild cognitive impairment (MCI) from healthy controls. This is a well-written paper containing interesting results, even though the idea of paper is very simple. However, several points need clarifying and certain statements require further justification, as shown in what follows.

(1)Is the selection of the K neighbors of each node independent? In other words, is the BFN a digraph?
(2)Can the direct calculation of the Euclidean distance between BOLD signals be seen as the distance between ROIs?
(3)More details about the dataset should be provided. The dataset contains different clinical information may cause potential bias for the study. In fact, the author should give a detailed explanation of Table 1 in the Data Acquisition and Preprocessing section. It is also recommended to provide a more detailed description of the preprocessing or links to relevant official websites.
(4)The present manuscript uses the ROIs of the AAL parcellation as nodes of the BFNs. However, AAL ROIs have been shown to suffer from low consistency: voxels inside of a ROI behave differently in time, leading to possible data losses when averaging the voxel time series. Therefore, I recommend that the authors include the reasons for using this parcellation.
(5)The reference format is not uniform. For example, some journal titles are in italics while others are not.

Experimental design

The experimental design is rigorous, the discussion is sufficient, the experimental results are compelling, and the data analysis is reasonable. It would be even better if the authors could make their code public.

Validity of the findings

It can be seen from the experimental results that the proposed method not only improves the classification accuracy, but also finds the top discriminative features. However, I suggest they explain the discriminative features from the perspective of biomarkers.

Additional comments

Overall speaking, the idea of the paper is simple, and the authors can express their method, experiments and findings clearly. I recommend to accept the paper for publication if the author can address the comments carefully.

Reviewer 3 ·

Basic reporting

This paper proposed a novel diagnostic method of brain disease models to distinguish MCIs from HCs. I think there are still some problems that need to be improved. The detailed comments can be downloaded from the PDF file.

Experimental design

The detailed comments can be downloaded from the PDF file.

Validity of the findings

The detailed comments can be downloaded from the PDF file.

Additional comments

no comment

Annotated reviews are not available for download in order to protect the identity of reviewers who chose to remain anonymous.

---

## Round 0.2 · Minor Revisions

Some minor concerns remain.

·

Basic reporting

Thank you for your efforts in addressing the majority of the concerns previously highlighted in your submission. Your revisions have significantly improved the manuscript, making the arguments clearer and enhancing the overall coherence of the work. However, I would like to draw your attention to a particular point that requires correction to avoid potential confusion:

In your response with In line 262, what is the meaning of ‘the issues’, you referred to “the issues” as representing “samples.” It is important to clarify that the term “issues” is generally not used to denote “samples” within the scientific community. This could lead to misinterpretation of your intended meaning. Using “issues” in this context might confuse readers and detract from the clarity and precision expected in a scholarly article.

Overall, aside from this specific terminology issue, your responses to other queries were well-articulated and insightful.

Experimental design

In your response with In line 262, what is the meaning of ‘the issues’, you referred to “the issues” as representing “samples.” It is important to clarify that the term “issues” is generally not used to denote “samples” within the scientific community. This could lead to misinterpretation of your intended meaning. Using “issues” in this context might confuse readers and detract from the clarity and precision expected in a scholarly article.
Overall, aside from this specific terminology issue, your responses to other queries were well-articulated and insightful.

Validity of the findings

no comment.

Additional comments

No comments

Reviewer 3 ·

Basic reporting

The author has made numerous revisions in response to the reviewer's comments.

Experimental design

The image quality needs to be enhanced, as many of the current images are not clear.

What does "bold" in Table 4 indicate?

P-value still needs revise.

Validity of the findings

no comment

---

## Round 0.3 · Minor Revisions

There are several concerns remaining from Reviewer 2.

·

Basic reporting

The response is written well.

Experimental design

no comment

Validity of the findings

no comment

Reviewer 2 ·

Basic reporting

no comment

Experimental design

1, in section 5.2, the authors describe "For verifying the validity of the presented approach, we compared it with the most commonly used time series methods in deep learning. We chose classic BiGRU, BiLSTM, and Transformer for comparison.", however, in Talbe 4, there are only the results of BiGRU, BiLSTM, and Transformer. what is the purpose of this experement and what is is intended to prove?

Validity of the findings

no comments

Additional comments

1、The experiment is too simple, the comparison methods are mostly baseline methods, and it is recommended to compare with the latest methods.

Reviewer 3 ·

Basic reporting

I suggest to accept this paper.

Experimental design

no comment

Validity of the findings

no comment

Additional comments

no comment

---

## Round 0.4 · accepted · Accept

The manuscript can be accepted now, congratulations!

Reviewer 2 ·

Basic reporting

no comment

Experimental design

no comment

Validity of the findings

no comment